



# Recent observations of superimposed ice and snow ice on sea ice in the northwestern Weddell Sea

Stefanie Arndt[1], Christian Haas[1], Hanno Meyer[2], Ilka Peeken[1], Thomas Krumpen[1]

[1]Alfred-Wegener-Institut Helmholtz-Zentrum für Polar- und Meeresforschung, 27570 Bremerhaven, Germany

[2]Alfred-Wegener-Institut Helmholtz-Zentrum für Polar- und Meeresforschung, 14473 Potsdam, Germany

*Correspondence to*: Stefanie Arndt (stefanie.arndt@awi.de)

**Abstract.**

Recent low summer sea ice extent in the Weddell Sea raises questions about the contributions of dynamic and thermodynamic atmospheric and oceanic energy fluxes. The roles of snow, superimposed ice, and snow ice are particularly intriguing, as they

are sensitive indicators for changes in atmospheric forcing, and as they could trigger snow-albedo feedbacks that could accelerate ice melt. Here we present snow depth data and ice core observations of superimposed ice and snow ice collected in the northwestern Weddell Sea in late austral summer of 2019, supplemented by airborne ice thickness measurements. Texture, salinity, and oxygen isotope analyses showed mean thicknesses of superimposed and snow ice of $0.11 \pm 0.11$ m and $0.22 \pm 0.22$ m, respectively, or 3 to 54 % of total ice thickness. Mean snow depths ranged between $0.46 \pm 0.29$ m in the south to $0.05 \pm$

0.06 m in the north, with mean and modal, total ice thicknesses between $4.12 \pm 1.87$ m to $1.62 \pm 1.05$ m, and 3.9 m to 0.9 m, respectively. These snow and ice properties are similar to results from previous studies, suggesting that the ice's summer surface energy balance and related seasonal transition of snow properties have changed little in past decades. This is supported by our additional analyses of the summer energy balance using atmospheric reanalysis data, and melt onset observations from satellite scatterometry showing little recent changes.

**1 Introduction**

After more than three decades of highly variable, but slowly increasing Antarctic summer sea ice coverage, ice extent has strongly declined between austral summer of 2016/2017 and 2018/2019, most notably in the Weddell Sea (Parkinson, 2019; Turner et al., 2020). In the northwestern Weddell Sea, a minimum for the last decade was observed in February 2019 (see Fig. 5 a below). A number of studies have related these negative ice extent anomalies to various atmospheric and oceanic processes,

often related to large, negative anomalies of the Southern Annual Mode (SAM), corresponding increases of cyclonic activity, and warm anomalies of several degrees Celsius at the Antarctic Peninsula and over the Weddell Sea (e.g., Francis et al., 2020; Schlosser et al., 2018; Turner et al., 2020; Wang et al., 2019). Together, these studies concluded that these processes contributed to warmer air, increased cloud-radiation feedbacks, increased liquid precipitation, and increased turbulent heat fluxes into the ice (e.g., Francis et al., 2020). In addition, earlier seasonal sea ice retreat, anomalous surface winds, and strong





atmosphere-ocean coupling caused stronger heat absorption by the upper ocean, more southward Ekman transport of warmer surface waters, and thus the warming of the Ocean Mixed Layer, amplifying the sea-ice loss (Meehl et al., 2019; Turner et al., 2020). Although these studies suggest a significant change in the coupled atmosphere/ice/ocean system in the Weddell Sea, the quantitative contribution of the individual components is not yet clear, nor how they might affect seasonal dynamic and thermodynamic sea ice properties.

The observed sea ice retreat could be caused by changes in atmospheric or oceanic thermodynamic forcing, or by wind and current related, dynamic changes of ice advection and deformation, and the processes are difficult to distinguish remotely. However, Antarctic sea ice is characterized by the occurrence of two types of snow-derived ice, namely superimposed ice and snow ice, whose presence and amounts are closely linked to the surface energy balance and ice and snow thicknesses, and who are therefore sensitive to changes of atmospheric heat flux towards the ice.

Superimposed ice forms during austral spring and summer when the snow cover becomes highly metamorphic during extensive thaw-refreeze cycles, and when internal snow melt water percolates to the colder snow/ice interface where it refreezes (Haas et al., 2001; Ackley et al., 2008; Kawamura et al., 2004; Nicolaus et al., 2003). Superimposed ice therefore forms on top of the sea ice and is characterized by large-grained, polygonal crystals, negligible salinity, and low $\delta^{18}$O oxygen isotope composition. In contrast to the Arctic, where summer snow melt is rapid and triggers snow albedo feedbacks and melt pond

formation (e.g., Webster et al., 2015), in the Southern Ocean thaw-refreeze cycles used to be the dominant form of surface melt (Arndt and Haas, 2019). This is due to the effects of dryer and cooler air which reduces the influence of clouds, and leads to more heat loss by stronger sensible heat fluxes and longwave radiation cooling (Andreas and Ackley, 1982; Nicolaus et al., 2006; Nicolaus et al., 2009; Vihma et al., 2009). The absence of strong surface melt allows survival of a year-round snow cover. It is unclear if the reported increases in atmospheric heat flux since 2016 (see above) have also increased snow melt and

superimposed ice formation, or have even led to the appearance of melt ponds on sea ice in the northwestern Weddell Sea.
In contrast, snow ice mostly forms during winter when seawater flooding of the snow/ice interface occurs where the ice has negative freeboard, i.e. when its surface is below the water level. This occurs when the snow depth approaches or exceeds approximately one-third of the ice thickness, which is frequently observed on Antarctic sea ice due to its relatively thin ice and typically thick snow cover (Eicken et al., 1994; Jeffries et al., 2001; Jeffries et al., 1997; Tian et al., 2020). Snow ice thus

forms from the refreezing of seawater-soaked snow, and is therefore fine-grained and saline, and has $\delta^{18}$O oxygen isotope concentrations between those of superimposed ice and sea ice (e.g., Eicken, 1998; Granskog et al., 2017). Given the dependence of snow ice on ice thickness and snow depth, long-term changes of its amount could indicate increases of snow accumulation causing more frequent flooding and less ice growth, or thinner ice from warmer air or increased ocean heat flux, or both (Ledley, 1991; Eicken et al., 1995; Powell et al., 2005).

Based on the above, observations of the amounts of superimposed ice and snow ice and their long-term changes can provide invaluable information on the underlying processes and atmospheric and oceanic contributions leading to changes of Antarctic sea ice mass balance and extent. Therefore, in order to evaluate changes in the amounts of superimposed ice and snow ice compared to previous studies in the same region, we conducted an intensive snow and ice sampling campaign in the





northwestern Weddell Sea in late summer 2019. In this study we present results based on ice core texture, salinity, and oxygen isotope analyses, and place them into broader context by means of ice thickness and snow depth measurements, thermodynamic modeling, and by backtracking of sampled floes to retrieve their age and origin. We support our ensuing discussion of decadal changes by means of analyses of the surface energy balance from reanalysis data and melt onset dates from satellite scatterometry.

## 2 Materials and methods

### 2.1 Study area and on-site measurements

The data and samples of this study were collected during the interdisciplinary Weddell Sea Ice (WedIce) project on board the German icebreaker *R/V Polarstern's* cruise PS118 (Haas et al., 2019) in the northwestern Weddell Sea in February and March 2019, i.e. at the end of the summer ablation period. Here, 14 ice floes were visited by helicopter and sampled for several hours (Fig. 1, left). At level, virtually representative locations of each floe, physical and biogeochemical ice properties were studied

by comprehensive sampling of up to 10 ice cores while snow properties were measured in vertical profiles by traditional snow pits (as described in Arndt and Paul, 2018). In order to better characterize the sampled floes and their significant variability, total (snow plus ice) thickness and snow depth were measured along 400 to 1200 m long transects across the entire floes with a ground-based multifrequency electromagnetic induction instrument (GEM-2, Geophex Ltd., Hunkeler et al. (2016)) and a GPS-equipped Magna Probe (Snow Hydro, Fairbanks, AK, USA). Finally, observations were placed into a regional context

by means of helicopter-borne electromagnetic (HEM) ice thickness surveys (Haas et al., 2009; Haas et al., 2008) with a total length of more than 2400 km covering the entire study region (Fig. 1, left). An overview of all ice stations is provided in the Appendix Table B1.



**Figure 1:** (left) Overview map of all sampled ice stations (red dots) and helicopter-borne electromagnetic induction sounding surveys (HEM, yellow lines). The ice station names are composed of the respective date and a consecutive station number. The dashed lines indicate the ice edge at the first day (February 22, 2019, blue) and last day (March 22, 2019, yellow) of our study. Background: Copernicus Sentinel-1 SAR images from March 20/21, 2019. (upper right) Back trajectories (black dots) of the sampled ice stations (red dots) during WedIce. The green line denotes the drift trajectory of the Snow Buoy 2018S59 deployed in February 2018 off the Ronne Shelf Ice. (lower right) Ice-age classification of all sampled ice cores: Second Year Ice (yellow), First Year Ice (blue), Young First Year Ice (red).

## 2.2 Ice core sampling and analysis

For this study, a total of 21 ice cores with a diameter of 0.09 m were taken from 14 ice stations, with 5 cores covering the entire ice column and the remaining ones just surface cores of at least the upper 50 cm of the ice. In the cold laboratory on board, detailed analyses of the ice crystal texture were performed on vertical thick sections between crossed polarizers (Lange, 1988). Based on the ice texture, all sampled ice cores were sliced into sections up to 15 cm and melted for the following



analysis of vertical salt (on board) and isotope profiles (in laboratory back home). Salinities were determined with a conductivity meter (Pocket conductivity meter WTW 3110) with a stated accuracy of 0.5% for each measurement. The melted samples were poured into sampling vials that were filled completely and tightly sealed. The vials were shipped at +4°C to the AWI ISOLAB Facility in Potsdam, where they were analyzed for stable water isotopes with Finnigan MAT Delta-S mass spectrometers using equilibrations techniques. The oxygen isotope composition is given as per mil difference relative to V-SMOW (‰, Vienna Standard Mean Ocean Water), with an internal $1\sigma$ error better than 0.1‰ for $\delta^{18}O$ (Meyer et al., 2000).

The correlation in the co-isotope plot (Fig. C1) suggests that the isotope data could be explained by a simple two-component mixture of snow and sea water, with little variation of the isotope fractionation during freezing. Snow/firn from Union Glacier region in the Weddell Sea sector has the respective endmember specifications (low $\delta^{18}O$ of -30-35 ‰ clearly under the Global Meteoric Water Line (GMWL); Hoffmann et al. (2020)).

Here we used the combination of salinity and oxygen isotopic composition to determine the respective fractions of superimposed ice and snow ice: All salt-free ice with an oxygen isotope composition smaller than 0.35 ‰ was classified as superimposed ice, and all salty ice with an oxygen isotope concentration $\delta^{18}O$ of up to 0.35 ‰ as snow ice following Eicken et al. (1994) and Schlosser et al. (1990) (Fig. 4).

## 2.3 Ice age classification

In order to interpret our results, it is important to know the type and age of the sampled ice which cannot easily be retrieved from the salinity or $\delta^{18}O$ profiles or other ice properties. Therefore we reconstructed the previous drift tracks of all sampled floes by means of Lagrangian sea-ice backtracking based on satellite-derived sea-ice motion fields (Krumpen et al., 2019). As the backtracking algorithm only works for ice concentrations above 20%, the drift tracks could only be determined for ice stations 2-11, all showing very similar tracks over their drift period of longer than one year since their initial formation near the Ronne Ice Shelf (Fig. 1, upper right). We have high confidence in this result as the drift tracks also closely agree with the drift of a buoy deployed in February 2018 off the Ronne Ice Shelf (Fig. 1, upper right; (Haas et al., 2019). To further constrain the age of the ice, we calculated the potential ice growth along these drift tracks with a simple one-dimensional thermodynamic sea-ice model. The model was forced by surface temperature, heat fluxes and snow fall from ERA5 reanalysis data (Copernicus Climate Change Service, 2017) (Appendix A). The resulting mean, potential ice thickness ($I_{pot}$) along all drift tracks was 1.53 ± 0.07 m overlain by 0.32 ± 0.01 m of snow. Based on these results, ice cores with a length of 1.53 ± 0.07 m were classified as first-year ice, thicker ice cores as second-year ice, and thinner ice cores as young first-year ice (Fig. 1, lower right). The classification considers the co-existence of different ice types in the study region which typically form by the refreezing of leads in the predominantly divergent ice conditions in the Weddell Sea (Haas et al., 2008).



## 2.4 Decadal time series of sea ice extent, melt onset, and surface energy balance

In order to support our discussion of long-term changes of superimposed ice and snow ice, we computed time series of February sea ice extent since 1979 from sea ice concentration data derived from the Nimbus-7 SMMR and DMSP SSM/I-SSMIS passive microwave sensors provided by the US National Snow and Ice Data Center (Cavalieri et al., 1996). Ice extent was calculated

for the complete Southern Ocean as well as for the western Weddell Sea west of 30°W (Fig. 5).

In addition, we updated our time series of snow melt onset dates since 1993 from satellite radar scatterometer data (Arndt and Haas, 2019) provided by the Scatterometer Climate Record Pathfinder (SCP) project at Brigham Young University (Long et al., 1993). Melt onset is detected from sudden changes and eventually increases of radar backscatter due to the appearance of thaw-freeze cycles and associated snow metamorphism and superimposed ice formation during the spring/summer transition

(Arndt and Haas, 2019; Haas, 2001). For this study we have computed the average melt onset for the available positions north of 69°S (location 1-3 in Arndt and Haas (2019)), representing the average position of the surveyed ice between the beginning of the melt season in late November and our sampling in February and March.

Finally, as superimposed ice formation and the survival of snow strongly depend on the surface energy balance, we computed time series of monthly net short- and longwave radiation as well as sensible and latent heat flux since 1979 from ERA5

reanalysis data (Copernicus Climate Change Service, 2017); the same data that were used to force the thermodynamic model in Section 2.3. The different energy flux components were added to derive the surface energy budget. We integrated the energy budget for the months December to February to obtain the surface energy budget during the summers of each year. Like melt onset, energy budget was computed for a region north of 69°S which approximately corresponds to the region the ice drifted through between December and February.


## 3 Results

### 3.1 Sea-ice and snow conditions

In February and March 2019, the northwestern Weddell Sea was characterized by the presence of at least three different ice regimes known from previous studies (Haas et al., 2008) and visible in satellite synthetic-aperture radar (SAR) imagery (Fig.

1): I. heavily deformed ice near the coast of the Antarctic Peninsula and along the Larsen Ice Shelf, II. east of that, a band of younger, thinner, and less deformed ice originating from the Ronne Ice Shelf, and III. in the very east, older, strongly deformed, thick ice originating from the southeastern Weddell Sea (Filchner Ice Shelf). These general ice regimes were interspersed with patches of thinner first-year ice originating from refrozen leads or sheltered areas at the Antarctic Peninsula. Based on visual interpretation of SAR images and the results of the thermodynamic model (Section 2.3) only one ice station was classified as

pure second-year ice, five as first-year ice, and six ice stations as young first-year ice. For another three stations, sampled cores were quite variable and classified with different ice ages. Independent of ice age, all ice core holes possessed a positive freeboard between 0.01 and 0.71 m (Fig. 2 c). Airborne sea ice thickness measurements revealed strong latitudinal gradients, with modal, total thicknesses between 3.9 m in the south and 0.9 m in the north, and mean values ranging from $4.12 \pm 1.87$ m to $1.62 \pm 1.05$ m, respectively (Fig. 2 a). Similar results were obtained by the GEM measurements with a mean, total ice


thickness of 4.08 ± 2.03 m at the southernmost ice station (PS118_20190226_4) and 1.50 ± 0.48 m in the north (PS118_20190313_10; Fig. 2 a). Similarly, mean snow depth ranged from 0.46 ± 0.29 m in the south to 0.05 ± 0.06 m in the north (Fig. 2 b). Although we occasionally observed floes with patches of bluish, bare ice probably originating from coastal fast ice in the Larsen region, we did not observe any melt ponds with characteristics known from the Arctic.

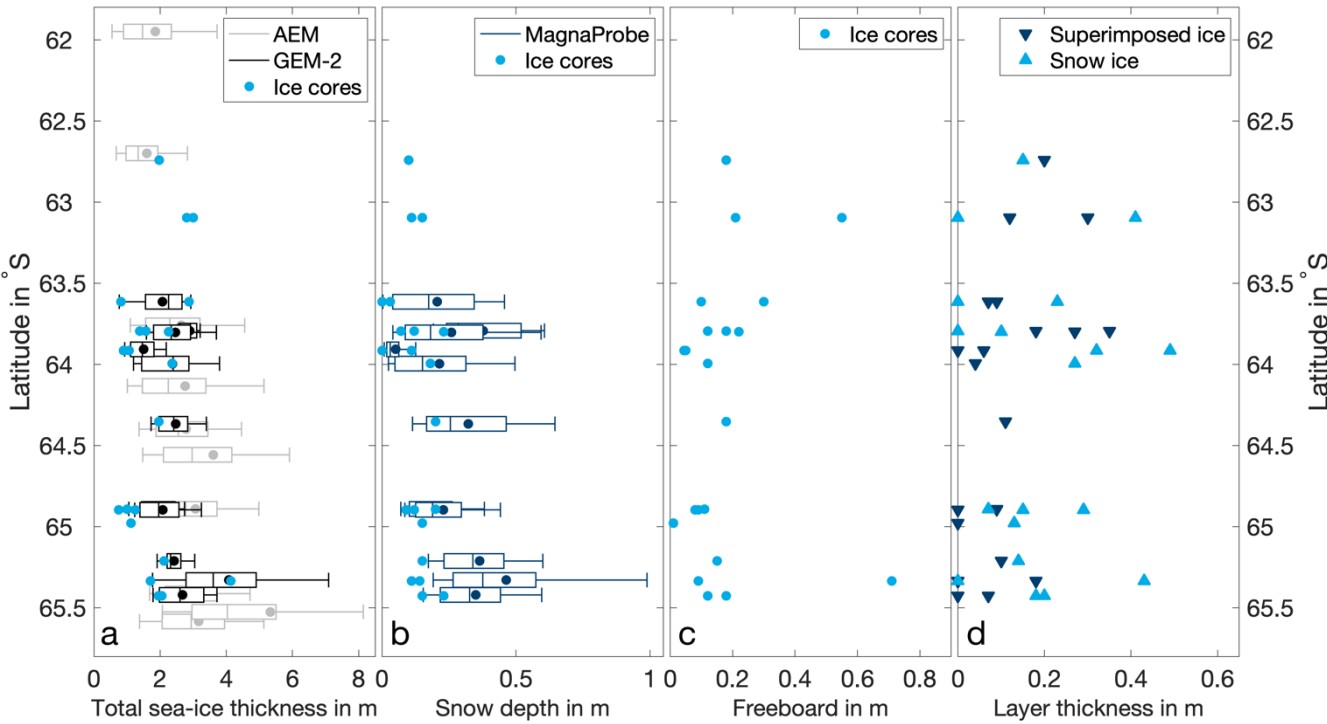

**Figure 2:** Ice and snow properties versus latitude: a. Total sea ice thickness (SIT, sea-ice thickness plus snow depth) derived from airborne EM sounding (AEM, grey), ground-based EM sounding (GEM-2, black) and ice cores (light blue); b. Snow depth measured with the MagnaProbe along GEM transects (dark blue) and measured at the ice-coring sites (light blue); c. Freeboard measured at the ice-coring sites (light blue); and d. Thicknesses of superimposed ice (dark blue triangles) and snow ice (light blue triangles) from all ice cores. Boxes are the first and third quartiles; whiskers the 10th and 90th percentile. Circles indicate means, horizontal lines in the box's medians.

### 3.2 Snow melt forms and superimposed ice

Based on detailed analyses of 24 snow pits, melt-freeze forms (Fierz et al., 2009) were identified as the dominant snow grain type with an average relative proportion of 60% (Fig. 3 c). These are typically caused by an early beginning and frequent recurrence of thaw-freeze cycles in the northwestern Weddell Sea (Arndt et al., 2016; Arndt and Haas, 2019). At all sites, the large-grained granular texture of snow often transitioned into the polygonal granular texture of underlying superimposed ice,





ranging from 0.05 to 0.24 m in thickness (Fig. 3 a). Within the superimposed ice, the grain size of polygonal grains often decreased downwards and transitioned into an underlying layer of finer grained orbicular granular ice, which could have led
to underestimation of the actual thickness and fraction of the superimposed ice. We therefore used the additional combined analysis of salinity and oxygen isotopes of all ice cores to clearly obtain superimposed ice thickness (Fig. 4). The fraction of superimposed ice in the total ice core length is about the same at 5-6 %, independent of the ice age (Fig. 3 d). The thickness of superimposed ice increased northwards, exceeding 0.3 m for the northernmost ice stations, while stations further south tended to have superimposed ice less than 0.1 m thick. (Fig. 2 d). Also, the median δ¹⁸O of superimposed ice gradually
decreased with increasing ice age from -14.9 ‰ for yFYI to -16.2 ‰ for FYI to -17.4 ‰ for SYI, supporting our classification approach as older ice is expected to originate from farther south with δ¹⁸O values of snow decreasing with increasing latitude (Eicken et al., 1994). Thus, the variability in the δ¹⁸O values is less ice type-specific but is subject to the ice floe origin and drift pattern.

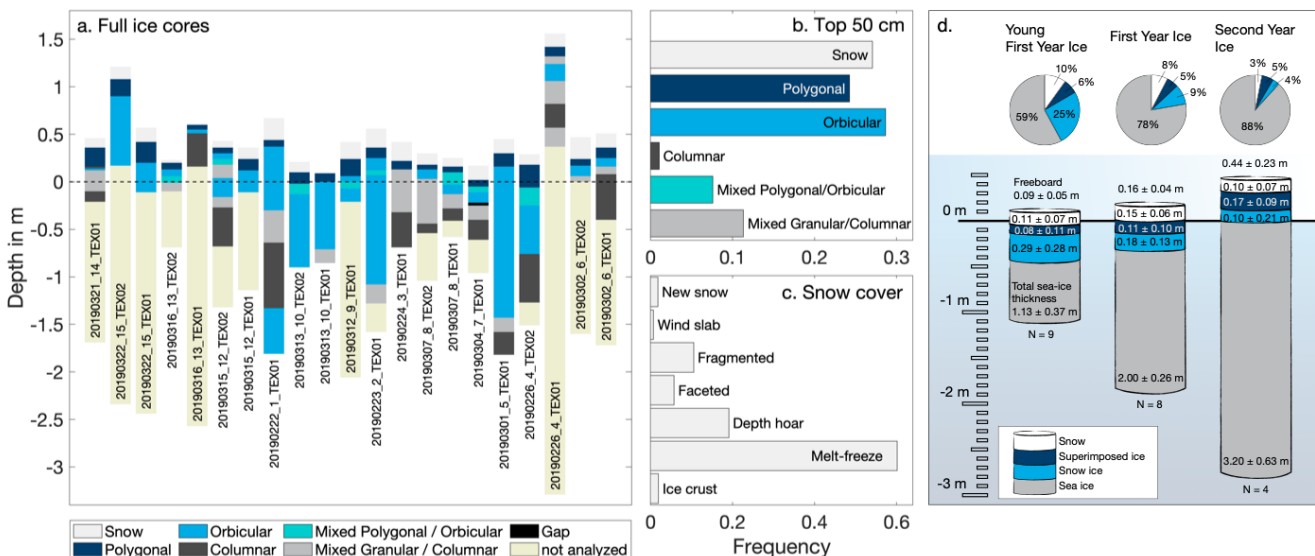

**Figure 3:** a. Overview of all sampled ice cores colored with the respective crystal texture. Ice cores are ordered according to their sampling location from north (left) to south (right, see Fig. 1). Ice core tags are composed of the respective ice station and the ice texture (TEX) core number. Beige-colored parts were not analyzed for texture. Ice core freeboard and draft are plotted relative to the water level (z = 0 cm, dotted black line). b. Relative frequency distribution of ice texture classes of the
00   surface ice cores. Includes only cores with more than 50 cm of ice texture information (15 out of 21 ice cores). c. Relative frequency distribution of snow grain types in all snow pits. d. Summary of average relative and absolute proportions of snow, superimposed ice and snow ice in total sea ice thickness (sea ice thickness plus snow depth) of all sampled young first year, first year and second year ice cores. Averaged ice cores are plotted relative to the water level (z = 0 cm, black line).

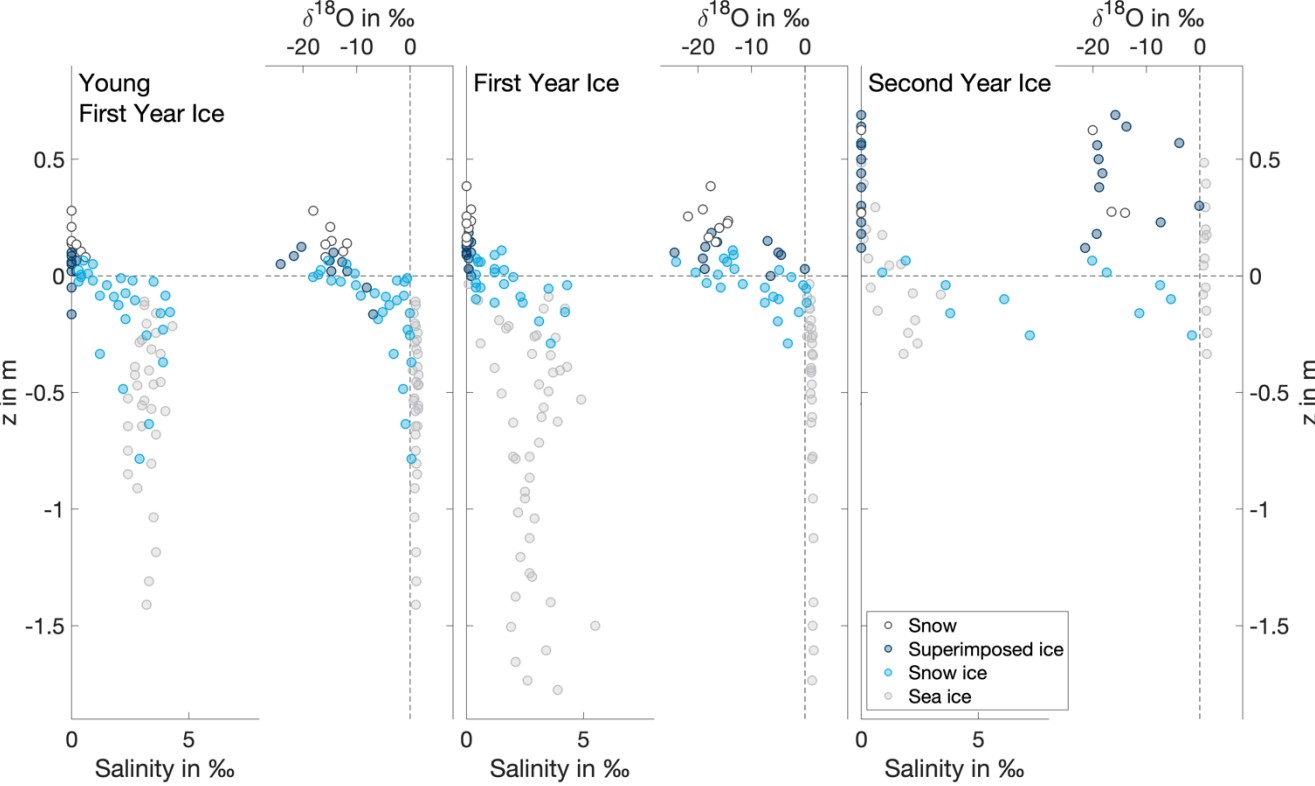

**Figure 4:** Vertical profiles of salinity and δ¹⁸O of all sampled ice cores from (left) young first-year ice, (middle) first-year ice, and (right) second-year ice. Markers are plotted at the center position of the individual core segment, relative to the water level (z = 0 cm). Colored circles mark snow (filled white circles), superimposed ice (filled dark blue circles), snow ice (filled light blue circles), and sea ice (filled light grey circles).

### 3.3 Fraction of snow ice

Beneath the salt-free layer of superimposed ice most cores possessed a salty layer of fine-grained, orbicular granular ice with salinities of up to 7.2 ‰. In accordance with salinity, also δ¹⁸O values increased to a median of -6.0 ‰. (Fig. 4). The excellent

correlation in the co-isotope plot (see Fig. C1) suggests that the isotope data could be explained by a simple two-component mixture of snow and sea water, with little variation of the isotope fractionation during freezing. Therefore, we interpreted these layers as snow ice. While the fraction of snow ice showed no latitudinal gradient, its contribution to the total ice core length varied considerably for yFYI (25%), FYI (9%) and SYI (4%) (Fig. 3 d). The fraction of both, snow ice and superimposed ice, i.e., meteoric ice, adds up to 3 to 54 % of total ice thickness.





20 Note that the spread of $\delta^{18}$O values of saline ice classified as snow ice was quite large, occasionally as low as -20 ‰ particularly near the snow/ice interface. Such low values suggest that meltwater may have contributed to the snow ice formation, e.g. during the period of melt onset. On the other hand, relatively high values near 0.35 ‰ may result from redistribution of $\delta^{18}$O isotopes by post-genetic changes in the ice, e.g. due to brine migration (Maksym and Jeffries, 2001). These processes can bias the interpretation of snow ice fraction in the cores. An analysis of the snow fraction of snow ice is beyond the scope of our

25 study (e.g., Eicken et al., 1994).

## 4 Discussion





**Figure 5**: Time series of ice extent, melt onset, and surface energy balance in the northwestern Weddell Sea (Section 2.4). (a) Sea ice extent (SIE) in the western Weddell Sea west of 30°W (wWS, solid line), and in the whole Southern Ocean (SO, dashed line), for February 1979-2020. Line shows linear fit to Weddell Sea data with a slope of +7 $10^{-3}$ Mio km$^2$ per year. (b) Snow melt onset north of 69°S from scatterometer data. (c) Surface energy budget north of 69°S (black line) and its individual components (colored lines). (d) Annual total summer surface energy budget (sum from December to February). The dashed line shows the linear fit with a slope of -0.7 Wm$^{-2}$ per year. The red box highlights the year of the WedIce expedition, 2019; blue boxes show years of field observations in 1997 and 2004/05 (ISPOL). Note that the year dashes on the x-axis represent January 01 of the respective year.

The observed, sudden sea ice retreat in the Weddell Sea in summer (Fig. 5 a) raises questions about the underlying causes and how they are related to other changes in ice and snow properties. Did atmospheric and oceanic processes also change the ice surface energy balance, cause increased snow melt and superimposed ice formation, and lead to changed proportions of snow ice? Given the small number of 21 ice cores from 14 stations, and the large regional variability and latitudinal gradients observed by us, this question is difficult to answer with certainty. However, our results are still the best recent observations available from the northwestern Weddell Sea to date, and other previous studies were subject to similar limitations.

Our ice thickness observations are similar to results from the Ice Station POLarstern (ISPOL) carried out in December 2004 in a region about 400 km farther south, where mean ice thicknesses of 3.01 ± 1.09 m were observed, with modal thicknesses of FYI and SYI of 1.2-1.3 m and 2.4-2.9 m respectively (Haas et al., 2008). Similarly, Lange and Eicken (1991) found mean total ice thicknesses from 0.9 ± 0.47 m to 3.11 ± 0.59 m for different ice classes in the northwestern Weddell Sea by the end of the 1980s. These results suggest that ice thicknesses in the northwestern Weddell Sea have changed little since then. The same applies to the snow depth distribution, as, e.g., Lange and Eicken (1991) have previously measured snow depths from 0.15 ± 0.59 m to 0.58 ± 0.07 m, and thus in the same range as our results in 2019 (Fig. 2).

Corresponding to little changed ice and snow thicknesses we did not observe unusual amounts of snow ice either, given that Eicken et al. (1994) reported similar proportions of 5 % in sea ice in the northwestern Weddell Sea in 1989 and 1992. As our coring and thickness drilling did not observe any negative freeboard, it is unlikely that much snow ice will form in the northwestern Weddell Sea at all, and that a very high snow accumulation is required to suppress the ice below the water level in this region. However, the snow ice observed in the northwestern Weddell Sea has formed in the preceding winter in regions farther to the south (Fig. 1, upper right), and therefore our results provide also some evidence of little changed ice growth and precipitation conditions there.

More importantly, via the frequency, intensity, and duration of snow melt events the amounts of superimposed ice are closely related to atmospheric energy fluxes. However, the mean superimposed ice thickness of 0.11 ± 0.11 m observed in February/March 2019 is similar to the 0.08 ± 0.06 m found by Haas et al. (2001) in February 1997 in the same region. It is





slightly larger than the 0.05 to 0.10 m found by Nicolaus et al. (2009) and Tison et al. (2008) in December 2004 in the ISPOL region slightly earlier and further south, and in a year with no exceptional February ice extent (Fig. 5 a).

In order to interpret the similar amounts of superimposed ice observed in 1997, 2004/05, and 2019, Fig. 5 b-d shows time series of melt onset and atmospheric energy fluxes. Melt onset occurred on Dec 31 in 1996/97 and 2004/05, and Dec 27 in 2018/19. Overall, there is no trend towards earlier melt onset, and there is little difference since 2016/17 when sea ice extent in the Southern Ocean plummeted. However, the total summer energy budget (Fig. 5 d) is a more relevant figure as it represents the accumulated energy that would have been available for snow thawing. It amounted to 170 Wm$^{-2}$ in 1996/97, 141 Wm$^{-2}$ in 2004/05, and 96 Wm$^{-2}$ in 2018/19, small differences which may explain why the amounts of superimposed ice changed little. We also note that air temperatures near the Antarctic Peninsula and the northwestern Weddell Sea were below average in January and February 2019 (Clem et al., 2020). Note that a more thorough analysis and modeling of energy fluxes and their impact on interannual variations of snow melt and superimposed ice formation (e.g., Nicolaus et al., 2006) is beyond the scope of this study.

In addition, Fig. 5 d shows that the surface energy balance in the northwestern Weddell Sea has decreased, by -0.7 Wm$^{-2}$ per year since 1979. While there is some correlation between low sea ice extent and high surface energy flux in extreme years like 1992/1993 and 2001/2002, there are little changes before and after 2016/17.

Based on these results, we conclude that we did not find any clear evidence for significantly increased atmospheric heat fluxes that would have changed the intensity of snow thaw and superimposed ice formation at the top of the sea ice. It is unlikely that increased melt from the top has contributed to the low sea ice extent in the northwestern Weddell Sea in the summer of 2019. This supports other studies which attribute the low sea ice extents to dynamic or oceanic processes rather than to thermodynamic atmospheric effects (Meehl et al., 2019; Reid and Massom, 2014; Reid et al., 2019; Reid et al., 2018).

Our conclusions are based on the assumption that atmospheric processes affecting sea ice melt and disappearance in the northwestern Weddell Sea also impact the surface of the surviving ice farther south. Obviously, it is impossible to assess melting processes in the formerly ice-covered open water regions when observations during the melting season for this area are not available. However, as gradients of most atmospheric properties typically have scales of variability of many 100s of kilometers one can expect that extreme conditions in the formerly ice-covered regions would also have affected neighboring regions farther south. In order to account for larger scales of atmospheric variability and for the conditions experienced by the ice observed by us, we have therefore carried out our analyses of melt and energy fluxes over the full regions through which that ice has drifted since the onset of melt in late November (Section 2.4).

Interpretation of changes in snow and superimposed ice thicknesses is complicated by the fact that the latter forms from the former. Increasing thicknesses of superimposed ice in summer will lead to simultaneous reductions of snow depth. For example, assuming densities of snow and superimposed ice of 330 and 850 kg m$^{-3}$ (Cheng et al., 2003), the 0.11 ± 0.11 m of superimposed ice found here correspond to an original snow depth of 0.27 ± 0.27 m. This conversion has an interesting side effect: given the thermal conductivity of snow of ≈ 0.3 W m$^{-1}$ K$^{-1}$, and of superimposed ice of ≈ 2.2 W m$^{-1}$ K$^{-1}$, the conversion of insulating snow to more conductive superimposed ice leads to a doubling of the effective thermal conductivity (Appendix



95 A, Eq. A4). This also doubles the potential conductive heat flux through the sea ice and snow, and therefore the potential sea-ice growth at the bottom of the ice in the following fall and winter (Appendix A, Eq. A7). Thus, increased bottom growth in winter may partially compensate for increased surface melt during summer, as long as the superimposed ice remains intact. In contrast, formation of superimposed ice in summer constitutes a downward heat flux where the latent heat released through meltwater refreezing contributes to the internal melting of the sea ice below and the formation of porous gap layers (Ackley

et al., 2008; Haas et al., 2001). The effects of gap layer formation on increased turbulent seawater flow through the ice and potential increased absorption of shortwave radiation by increased amounts of photosynthesizing microorganisms are difficult to predict.

## 5 Conclusions

In February/March 2019 we have carried out extensive snow depth and ice thickness measurements and sea ice core analyses in the northwestern Weddell Sea, to gain more insights into the possible impact of atmospheric and oceanic changes related to the strong sea ice retreat observed between 2016/17 and 2018/19. Our results showed similar sea ice properties as in the few previous studies carried out since the 1980s/90s, and in particular we did not find unusual amounts of meteoric ice, i.e. of superimposed ice or snow ice which are sensitive indicators of the sea ice surface energy balance and the relation between

snow depth and ice thickness, respectively. In fact, we showed that melt onset dates and the surface energy balance between 2016/17 and 2018/19 were similar to previous years. These results support other studies showing that the low sea ice coverage in the northwestern Weddell Sea in February 2019 must have been the consequence of dynamic or oceanic processes rather than of thermodynamic atmospheric effects, i.e., of advection by winds and currents and increased ocean heat rather than of increased air temperatures, turbulent fluxes, or longwave radiation.

We suggest that snow melt and metamorphism and superimposed ice formation as a are sensitive indicators for changes in the surface energy balance, and together with snow ice allow to distinguish between atmospheric and oceanic contributions to increased sea ice melt and retreat. Based on observations of Arctic snow ice, Granskog et al. (2017) suggested that with thinner sea ice and more snow ice there might be an "antarctification" of Arctic sea ice. Here we suggest that the potential "arctification" of Antarctic sea ice would imply an increase in surface melt, reduction of albedo due to metamorphic and wet

snow, and the eventual ice-albedo feedback supported appearance of melt ponds caused by more Arctic-like atmospheric conditions with warmer and moister air and increased surface heat and radiation fluxes. In this regard, superimposed ice plays an intermediate role, as initially more superimposed ice indicates more snow melt and snow-to-ice conversion. However, this could also contribute to more rapid sea ice melt underneath as its formation constitutes a downward heat flux (Haas et al., 2001; Ackley et al., 2008). In addition, once surface melt would reach Arctic levels the eventual disappearance of snow would

continue with the melting and disappearance of the underlying superimposed ice (Nicolaus et al., 2003). However, our results and the absence of melt ponds in the summer of 2019 imply that strong arctification of sea ice in the northwestern Weddell Sea has not yet commenced. This, and the large thickness of the sea ice in the region suggest that more substantial atmospheric and oceanic changes are required before perennial sea ice will completely disappear from the Weddell Sea. We strongly



encourage future sea ice studies in the northwestern Weddell Sea to include extensive observations of superimposed and snow ice and the processes of their formation to better document the ongoing, if subtle, changes of atmosphere-ice-ocean processes in the region. In addition, satellite methods to derive snow and ice thawing and melt remotely should be improved and expanded (e.g., Arndt and Haas, 2019; Arndt et al., 2016).





## Appendix A: One-dimensional thermodynamic sea-ice model

In order to calculate potential sea-ice growth rates along the drift trajectories of the sampled ice floes, we used a simple one-dimensional thermodynamical ice growth model based on the number of freezing degree days (Leppäranta, 1993), forced by surface heat fluxes and temperature as well as snow fall rates from ERA5 data (Copernicus Climate Change Service, 2017). The model is simplified by using constant salinity profiles and constant oceanic heat flux of 3 Wm$^{-2}$ (Robertson et al., 1995).

For the surface energy balance, it is assumed that the sea-ice surface is in thermal equilibrium with the atmosphere, requiring that the heat fluxes into the ice and out of the ice ($F_{s,net}$) are balanced, as given in the following equation:

$$F_{S,net} = (1 - \alpha)F_R + F_L^{\downarrow} - F_L^{\uparrow} + F_{sens} + F_{lat} - F_C \text{ (Equation A1)}$$

Here, the individual heat flux terms refer to

$\alpha$ : Surface albedo of snow/ice, given by ERA5 data;

$F_R$ : Surface solar radiation downwards, given by ERA5 data;

$F_L^{\downarrow}$ : Surface thermal radiation (longwave) downwards, given by ERA5 data;

$F_L^{\uparrow}$ : Outgoing longwave radiation, calculated by Stefan-Boltzmann law $F_L^{\uparrow} = \varepsilon T_{surf}^4$ **(Equation A2)**,

with $T_{surf}$ : Sea surface temperature, given by ERA5 data;

$F_{sens}$ : Surface sensible heat flux, given by ERA5;

$F_{lat}$ : Surface latent heat flux, given by ERA5;

$F_C$ : Conductive heat flux through snow and ice, $F_C = -\frac{T_{surf} - T_{bot}}{\frac{h_{ice}}{k_{ice}} + \frac{h_{snow}}{k_{snow}}}$ (assuming a linear temperature gradient, **Equation A3**),

with the so-called effective heat conductivity through snow and ice, $k_{eff} = \frac{h_{ice}}{k_{ice}} + \frac{h_{snow}}{k_{snow}}$ **(Equation A4)**,

leading to $F_C = -\frac{T_{surf} - T_{bot}}{k_{eff}}$,

with $T_{bot}$ : Bottom ice temperature, $T_{bot}$ = 271.35 K;

$h_{ice/snow}$ : Sea-ice and snow thickness;

$k_{ice}$ : Thermal conductivity of sea ice, $k_{ice}$ = 2.2 W m$^{-1}$ K$^{-1}$; and

$k_{snow}$ : Thermal conductivity of snow, $k_{snow}$ = 0.3 W m$^{-1}$ K$^{-1}$.

The net heat flux at the bottom of the ice, $F_{B,net}$, is simply given as the difference between the conductive heat flux, $F_C$, and the constant ocean heat flux, $F_O$ = 3 W m$^{-2}$,

$$F_{B,net} = F_C - F_O \text{ (Equation A5)}.$$

The resulting growth/melt rates at the surface and bottom of the ice floe dh/dt are finally calculated as a function of time. Assuming a vertical heat transfer only, leads to a surface melt/growth rate $dh_{surf}/dt$ of

$$\frac{dh_{surf}}{dt} = \frac{F_{S,net}}{\rho_{ice/snow} L} \text{ (Equation A6)},$$

and bottom melt/growth $dh_{bot}/dt$ of

$$\frac{dh_{bot}}{dt} = \frac{F_{B,net}}{\rho_{ice} L} \text{ (Equation A7)},$$



with

$\rho_{ice}$ : Density of ice, $\rho_{ice}$ = 910 kg m$^{-3}$;
$\rho_{snow}$ : Density of snow, $\rho_{snow}$ = 330 kg m$^{-3}$; and
L : Latent heat of fusion (freezing of water), L = 334 kJ kg$^{-1}$.

## Appendix B: Overview of all sampled ice stations

**Table B1:** List of all sea ice stations within the WedIce project during PS118. The ice station names are composed of the respective date and a consecutive station number. Abbreviations for the used gear at each station are given as following: SIT – Manual sea-ice thickness drillings, SPIT – Snow pit, SMP – SnowMicroPen, SDMP – Snow depth measured with Magna Probe, GEM – Sea-ice thickness measured with ground electromagnetic sounding device, CORE – Physical and biological ice coring, WATER – Surface water sampling.

| Station | Date | Latitude | Longitude | Gear |
|---|---|---|---|---|
| **PS118_20190222_1** | 2019-02-22 | 63°47.900'S | 56°19.220'W | SIT, SPIT, SMP, SDMP, GEM, CORE, WATER |
| **PS118_20190223_2** | 2019-02-23 | 64°21.183'S | 56°23.072'W | SIT, SPIT, SMP, SDMP, GEM, CORE, WATER |
| **PS118_20190224_3** | 2019-02-24 | 64°53.550'S | 57°14.513'W | SIT, SPIT, SMP, SDMP, GEM, CORE, WATER |
| **PS118_20190226_4** | 2019-02-26 | 65°20.136'S | 58°0.384'W | SIT, SPIT, SMP, SDMP, GEM, CORE, WATER |
| **PS118_20190301_5** | 2019-03-01 | 65°12.720'S | 57°34.793'W | SIT, SPIT, SMP, SDMP, GEM, CORE, WATER |
| **PS118_20190302_6** | 2019-03-02 | 65°25.548'S | 57°55.705'W | SIT, SPIT, SMP, SDMP, GEM, CORE, WATER |
| **PS118_20190304_7** | 2019-03-04 | 64°58.662'S | 57°40.782'W | SIT, SPIT, SMP, SDMP, CORE, WATER |
| **PS118_20190307_8** | 2019-03-07 | 64°53.814'S | 57°48.420'W | SIT, SPIT, SMP, SDMP, GEM, CORE, WATER |
| **PS118_20190312_9** | 2019-03-12 | 63°59.693'S | 55°36.379'W | SIT, SPIT, SMP, SDMP, GEM, CORE, WATER |
| **PS118_20190313_10** | 2019-03-13 | 63°54.895'S | 55°40.580'W | SIT, SPIT, SMP, SDMP, GEM, CORE, WATER |
| **PS118_20190314_11** | 2019-03-14 | 63°50.269'S | 55°39.951'W | SPIT |
| **PS118_20190315_12** | 2019-03-15 | 63°47.720'S | 55°27.708'W | SIT, SPIT, SDMP, GEM, CORE, WATER |
| **PS118_20190316_13** | 2019-03-16 | 63°36.852'S | 56°08.739'W | SIT, SPIT, SDMP, GEM, CORE, WATER |
| **PS118_20190321_14** | 2019-03-21 | 62°44.485'S | 53°03.065'W | CORE |
| **PS118_20190322_15** | 2019-03-22 | 63°05.700'S | 54°17.583'W | SIT, SPIT, CORE, WATER |




**Appendix C: Co-isotopic diagram**

90

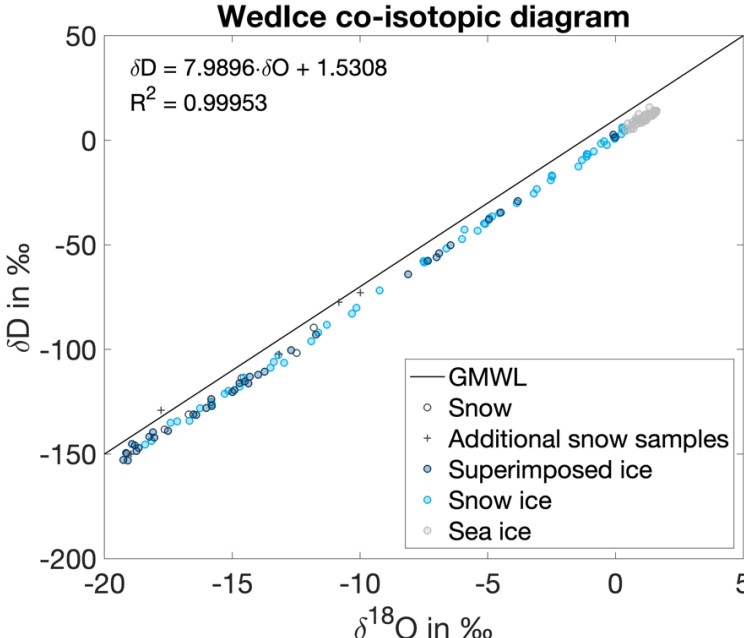

**Figure C1:** Oxygen isotope ($\delta^{18}$O)-hydrogen isotope ($\delta$D)-diagram for all WedIce samples differentiated by ice type. The Global Meteoric Water Line (GMWL; Craig (1961)) is given for comparison. During some ice stations additional snow samples have been taken next to the ice core sampling sites which were also analyzed for isotopes and are added here with
95  crosses (+).

**Data availability**

All ERA5 data from ECMWF are from accessed and downloaded from the Copernicus Climate Change Service (last access: April 12, 2021). Sea ice concentration data are from the NASA National Snow and Ice Data Center Distributed Active Archive Center, Boulder, Colorado, USA (last access: January 30, 2021). Satellite data of radar backscatter were kindly provided by
00  the Scatterometer Climate Record Pathfinder (SCP) project, sponsored by NASA (http://www.scp.byu.edu/, last access: January 28, 2021). All presented field data are achieved at PANGAEA: https://doi.pangaea.de/10.1594/PANGAEA.928948, https://doi.pangaea.de/10.1594/PANGAEA.928966, and https://doi.pangaea.de/10.1594/PANGAEA.929010.



## Author contribution

SA, CH and IP conducted the field work for the WedIce project. HM performed the isotope analysis in the laboratory in Potsdam. TK contributed the backtracked trajectories for the sampled ice floes. SA conducted all analysis for the paper and wrote it with contributions of all co-authors to the discussion and actual writing.

## Competing interests

The authors declare that they have no conflict of interest.

## Acknowledgements

We gratefully acknowledge the support of the cruise leader Boris Dorschel, all involved scientists, the helicopter team on board and the captain and crew of *R/V Polarstern* during expedition PS118 (grant number AWI_PS118_11). Special thanks go to Erika Allhusen and Kerstin Jerosch for the fantastic teamwork on the ice and in the laboratory. Also, we thank Mikaela Weiner for carrying out the stable isotope analyses in the AWI ISOLAB Facility, and Martin Werner for fruitful discussions on the isotope data. This work was supported by the German Research Council (DFG) in the framework of the priority program "Antarctic Research with comparative investigations in the Arctic ice areas" by grants SPP1158 and AR1236/1, and the Alfred-Wegener-Institut Helmholtz-Zentrum für Polar- und Meeresforschung.

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
