# Peer review of "Recent observations of superimposed ice and snow ice on sea ice in the northwestern Weddell Sea"

_The Cryosphere, 2021_

## Author Comment (AC1)

**Review response on**

**"Recent observations of superimposed ice and snow ice on sea ice in the northwestern Weddell Sea" by Stefanie Arndt et al.**

**Anonymous Referee #1**

This paper presents in situ observations of superimposed ice and snow ice in sea ice cores from the Weddell Sea, and attempts to place those results in the context of recent sea ice variability and the potential for this variability to be connected with changes in surface energy balance and melt, and if the superimposed ice and snow ice can provide an indicator of these changes. Data such as these are rare, particularly for superimposed ice, and as such are a valuable contribution and worth publishing in their own right, but the authors go further to place these data in context of ongoing sea ice variability. While, this analysis is necessarily fairly qualitative, it does add some interesting perspective on the larger scale sea ice variability. The paper is well written and figures well-presented and clear. I recommend publication with some fairly minor suggested changes.

Full disclosure - I was one of the reviewers of the orginal manuscript. I am not sure if I am the one who the authors suggest misunderstood their intent for how superimposed ice and snow ice serve as climate indicator. In any case, I did not recommend rejection. I state that here so that it is clear that I did not believe there were any fatal weakness. While I agreed with the authors that their ice core observations did not suggest any significant change in surface melt that may have resulted from changes in surface forcing, I recommended that they provide additional information on that forcing to corroborate this result, since, despite the many studies suggesting potential drivers for recent sea ice variability in this region, it was not clear that there was any evidence for atmospheric forcing that would have caused increased surface melt. i.e. should we have actually expected a change or not? They have now added this context and it supports the interpretation of the in situ data, so my concerns have been mostly addressed.

I do think you could still be more clear what you believe the superimposed ice suggests about sea ice and climate forcing variability. As your data and reanalysis show, there isn't really any difference among the three years you have in situ data in the forcing or melt onset, so why would you expect any change in surface ice properties? Since neither your data nor the forcing show any change, then it seems like speculation about possible arctification is unwarranted. Nothing wrong with negative results, but this framing could lead to possible confusion about what should we expect to be happening. What is interesting, though, is that there do appear to be trends in melt onset and SIE until 2016, when there is a dramatic decline in SIE, but not melt onset (nor in other surface forcing). As that is consistent with your in situ data, this all suggests that the drivers of this recent change were through the ocean. You do say this, so I agree with your interpretation. But I think some slight changes to the text (maybe just a few sentences) could help make this more clear what conclusions are supported by the data, and what are not. (see more on this in comments below)

We highly appreciate the great work that the reviewer put into reviewing our manuscript. This work is really excellent and we realize that he / she is really familiar with this field of research and has a great expertise to make most useful comments and suggestions. We thank the reviewer for the comprehensive comments on distinguishing between conclusions that support our data and those that do not, and the more explicit outline of parameters that show change and those that do not. Therefore, we added respective sentences and explanations in the manuscript, as descripted related to the detailed comments below.

**Detailed comments**

Note: line numbers did not print out in full on the pdf (first digit missing for 3-digit numbers), so I may have some of them wrong – I have indicated page numbers as well in some cases to help.

Line 45 – "used to be" implies it not longer is the case. But of course it probably still is the case. Change the wording here to something like "are".

The term "used to be" was meant to underline the fact that thaw-freeze cycles are so far the dominant surface melt process but that this may be changing in the future. We therefore changed the sentence to:

*In contrast to the Arctic, where summer snow melt is rapid and triggers snow albedo feedbacks and melt pond formation (e.g., Webster et al., 2015), in the Southern Ocean thaw-refreeze cycles are so far the dominant form of surface melt (Arndt and Haas, 2019).*

Line 60-69 – this is all fine, but a bit vague. This could be stronger if you provide a specific goal of the paper here. i.e., My impression is that your intent here is to compare 2019 observations with ones from earlier cruises to see if there is an indication of the impact of climate variability on sea ice surface properties, and if so (or even if not) whether those differences (or lack thereof) can provide any insight into the causes (or impacts) of recent sea ice variability. I think this speaks to what was unclear in the prior manuscript.

We thank the reviewer for the critical comment and adapted the sentence accordingly to make the objective clearer to the reader:

*Based on the above, observations of the amounts of superimposed ice and snow ice and their long-term changes can provide invaluable information on the underlying processes. Thus, the question arises whether there are indications of increased atmospheric heat fluxes leading to changes of Antarctic sea ice mass balance and extent.*

Line 74 – what is meant by "virtually representative" here? Do you mean something like "fairly representative"

"Virtually representative" is meant in that context as observations by eye (without measurements but just by looking on the floes from the helicopter). To make this point clearer, we changed the sentence to:

*At apparently level and representative locations of each floe, physical and biogeochemical ice properties were studied by comprehensive sampling of up to 10 ice cores while snow properties were measured in vertical profiles by traditional snow pits (as described in Arndt and Paul, 2018).*

Line 105-108 (page 5) – Did you have d18O for the snow as well? Would this not provide a better snow endmember, since that is the snow that gets incorporated into superimposed and snow ice? I guess you don't use this endmember to classify, though, so not really important.

As noted by the reviewer, we do not use the endmember for further analysis or classification and therefore have not added any data other than that from the ice cores.

Line 109-111 (page 5) – Can you give an estimate of what the uncertainty caused by these assumptions might be? For instance, the 0.35 value could vary depending on where it formed (if there is any variability in the water), and there are several processes which could change these values somewhat. It looks like it's not super important from Figure 4, but for snow ice you do have quite a few values close to 0 ppt, and a couple superimposed ice values right about 0 ppt. I have a hard time understanding how you can get superimposed ice with such a value. Even rain should give you a fairly negative value. It seems more likely that this is sea ice that has been flushed of brine in some way.

We agree with the reviewer that it's worth calculating the uncertainty based on the oxygen isotope concentration of 0 compared to the used 0.35 ‰. We therefore added a respective sentence with the calculated uncertainty of 14% in the results section (Section 3.3):

*To account for these biases, a mean uncertainty of the relative proportion of snow ice of 14% is derived, assuming an oxygen isotope concentration $\delta^{18}O$ of 0‰ compared to the threshold of 0.35‰ used to derive the respective fractions of snow ice and superimposed ice.*

Figure 3 – it is interesting that there is so little snow ice in the second year ice – in other regions, you get extensive flooding in spring summer, and if that survives into a second year, you would get lots of snow ice in the resulting second year ice. It appears this is not true here, which I suppose means that for the second year ice, it spent its first summer in the south and so was not exposed to too much precipitation and bottom melt, so maybe didn't flood too much in the summer? I suppose that you have positive freeboards here suggests flooding in summer isn't that big a deal. Do you have any indication that this might be limited by the surface snow melt and superimposed ice formation, so snow loads are kept low in spring? I suppose superimposed ice layer thickness are relatively modest, so it's maybe more the snow is just not very deep to begin with.

We agree with the reviewer that the low amount of snow ice, especially on second-year ice, is worth a mentioning. We therefore added a respective paragraph in the discussion section on the snow ice:

*Overall, the amount of snow ice on second-year ice is relatively low. This relates to the fact that the ice spent the first year mainly in the southern Weddell Sea as indicated by our snow buoy (Figure 1 top right). There, only limited snow fall occurs due to the strong continental high-pressure influence (Van Den Broeke and Van Lipzig, 2004).*

Line 252 (page 11) – Are you sure this is a correct comparison? I do not have a copy of Eicken et al (1994) handy, and it is not available online, but I believe this percent (4% from the abstract), is the fraction of snow in the core, and not the thickness of the snow ice layers – i.e. they reported both a snow fraction calculation and a layer thickness (which is how you define snow ice here). If that is correct, they would actually have significantly more snow ice comparing like with like.

We believe our statement is correct. Eicken et al. reported that "roughly 4% of the total ice thickness consist of meteoric ice (FY 3%, SY 5%)". I.e. it is the amount of ice per total thickness, like in our results.

Figure 5 and associated text in discussion and conclusions – There are several points here that are worth mentioning.

First, you do mention that the extreme years of high flux do correspond to low SIE. But not to all low SIE years (notably, the most recent ones). This suggests (unsurprisingly) that there are other processes that drive low sea ice extents (in fact the surface net shortwave increases in those years may simply be because of low SIE and you have some absorption by open ocean, as it is not clear if you screened the fluxes for over ice only). But, interestingly, surface melt onset does not necessarily correspond to years of higher fluxes. I'm not sure if there is anything more you can say about that for this study. But what is notable is that the three years you have superimposed ice data for all have fairly similar SIE, and snowmelt onset dates. So, really, the change from 2016-2019 is not so much an anomaly, but returning to normal (at least for the western Weddell; I think you still have an anomaly for the Weddell overall). So, why would you expect to see any difference? Same sea ice extent, same melt onset, fairly similar surface fluxes. Your text on this is fine, as you do acknowledge this. But it is worth pointing out that that does not mean that there have been no changes in surface melt. Over the long term, it appears ice extent has been increasing, and melt onset has been occurring later. I wonder what you might have seen if you went in 1992 or 2015? The former you have very early onset, high fluxes, and low extent, and the latter you have late onset, high extent, and low fluxes. So, if anything, if you were able to observe every year maybe you'd see an overall decrease in superimposed ice extent? What happened in recent years is interesting, and I agree this does corroborate those studies that suggest the role of ocean forcing. In that sense, this analysis, while necessarily just suggestive, is quite nice. Maybe just add a couple sentences highlighting the similarity and differences in the years with in situ data and those without.

We agree that the given points by the reviewer should be considered and mentioned clearer in the manuscript in order to not provide misleading conclusions to the reader. We therefore added the following sentences to the manuscript:

*However, our results are affected by some sampling bias due to the fact that we only have in-situ data from three years in which sea ice extent and snowmelt onset dates were quite similar (Figure 5). With increasing sea ice extent in the Weddell Sea overall, and melt onset tending to start later while surface fluxes show little change we may speculate that superimposed ice formation has decreased overall. In contrast, in 1992 for example, when melt onset was early, surface fluxes were high, and sea ice extent was low (Figure 5), superimposed ice formation may have been strong, and it may have been lowest in 2015 with a comparatively late melt onset, low surface fluxes, and the highest sea ice extent for the Weddell Sea (Figure 5).*

*In addition, Fig. 5 d shows that the surface energy balance in the northwestern Weddell Sea has decreased, by -0.7 Wm$^{-2}$ per year since 1979. While there is some correlation between low sea ice extent and high surface energy flux in extreme years like 1992/1993 and 2001/2002, there are little changes before and after 2016/17.* *However, the exact link between extremes in sea ice extent and the corresponding inverse extremes in surface energy fluxes are not yet fully understood.*

Lines 309-311 (page 13) – be careful here. As noted above, it is similar to previous years in which you have in situ obs, but there are differences with other years. I think you should acknowledge this lack of data here.

We agree with the reviewer to point out the lack of data in extreme years and therefore extended the sentence as following:

*In fact, we showed that melt onset dates and the surface energy balance between 2016/17 and 2018/19 were similar to previous years*, *however, data on extreme years as e.g., minimum/maximum sea ice extent or surface energy fluxes, are lacking.*

Lines 315-332 (page 13-14) – This is an interesting idea. But because you only have 3 years of data to infer this from, I think it is too bold to say it has not yet commenced. I suggest toning down the wording a wee bit, e.g. "However, our results and the absence of melt ponds snow no evidence for strong arctification of sea ice in the northwestern Weddell Sea in 2019".

We agree and adjusted the sentence as suggested.

*However, our results and the absence of melt ponds* *show no evidence for strong arctification of sea ice in the northwestern Weddell Sea in 2019.*

Appendix A and identification of FY vs SY ice. The text implies the distinction of ice types is based on the thickness model. But you could easily have thick first year ice if it rafter – or was it clear from the cores and the topography that major rafting and deformation was not an issue? I would think you'd have additional evidence from the ice core structure, salinity, etc to support whether the ice was FY or SY.

As mentioned in the manuscript, sampling sites are chosen from level ice only. Thus, both by observations by eye and by drilling the ice cores, we can be sure that we had no evidence of rafting or deformation.

---

## Author Response (AR1)

**Review response on**

**"Recent observations of superimposed ice and snow ice on sea ice in the northwestern Weddell Sea" by Stefanie Arndt et al.**

**Anonymous Referee #2**

As with Referee 1, I was also a previous reviewer of an earlier ms on this topic.

The paper is a good representation of the observations on superimposed and snow ice formation in the Western Weddell Sea. With the benefit of Reviewer 1's posted detailed comments, I have only a brief comments to add.

We thank the reviewer for the overall positive feedback and that he/she acknowledged the progress of the manuscript.

On the argument that the conditions for melting of the snowpack in the Weddell Sea that would lead to a triggering of ice albedo feedback and widespread melting (the 'arctification" of the Western Weddell Sea), they may want to add an additional sentence or two. The diurnal freeze-thaw cycle that exists in the Antarctic because of its lower latitude can shift the energy balance from positive shortwave to negative longwave especially during late Jan and Feb as the daylight hours shorten away from the solstice. Along with the deeper snow cover in the Weddell Sea, the triggering of ice-albedo feedback may be delayed by this diurnal radiation effect.

We agree with the reviewer that the latitudinal dependence of the radiation should be mentioned in the conclusion. However, a deeper analysis of this effect is beyond the scope of the manuscript. We therefore added the following paragraph to the conclusion section:

*When discussing the potential future "arctification" of Antarctic sea ice one also has to take into account the fundamentally different solar radiation conditions experienced by Arctic and Antarctic sea ice in summer due to their different latitudinal occurrence, with Arctic sea ice mostly residing at latitudes above 80° and Antarctic sea ice below 80° (e.g. Haas, 2003). This implies that Antarctic sea ice is subject to stronger diurnal cycles where nighttime longwave radiation cooling favors superimposed ice formation (Arndt and Haas, 2019) and may reduce the effectiveness of the ice-albedo feedback in leading to melt pond formation.*